# Exposure to Air Pollution in Rural Malawi: Impact of Cooking Methods on Blood Pressure and Peak Expiratory Flow

**DOI:** 10.3390/ijerph18147680

**Published:** 2021-07-20

**Authors:** Fiona Mabonga, Tara K. Beattie, Kondwani Luwe, Tracy Morse, Caitlin Hope, Iain J. Beverland

**Affiliations:** 1Department of Civil and Environmental Engineering, University of Strathclyde, James Weir Building, 75 Montrose Street, Glasgow G1 1XJ, UK; fiona.sutherland@strath.ac.uk (F.M.); t.k.beattie@strath.ac.uk (T.K.B.); tracy.thomson@strath.ac.uk (T.M.); caitlin.hope@strath.ac.uk (C.H.); 2Centre for Water, Sanitation, Health and Appropriate Technology Development (WASHTED), University of Malawi (Polytechnic), Blantyre 34310, Malawi; kondwaluwe@gmail.com

**Keywords:** PM_2.5_, exposure, cookstove, indoor, outdoor

## Abstract

We made static and personal PM_2.5_ measurements with a miniature monitor (RTI MicroPEM) to characterise the exposure of women cooking with wood and charcoal in indoor and outdoor locations in rural Malawi, together with measurements of blood pressure and peak expiratory flow rate (PEFR). Mean PM_2.5_ concentrations of 1338 and 31 µg/m^3^ were observed 1 m from cookstove locations during cooking with wood and charcoal, respectively. Similarly, mean personal PM_2.5_ exposures of 706 and 94 µg/m^3^ were observed during cooking with wood and charcoal, respectively. Personal exposures to PM_2.5_ in indoor locations were 3.3 and 1.7 times greater than exposures observed in equivalent outdoor locations for wood and charcoal, respectively. Prior to the measured exposure, six out of eight participants had PEFR observations below 80% of their expected (age and height) standardised PEFR. We observed reductions in PEFR for participants cooking with wood in indoor locations. Five out of eight participants reported breathing difficulties, coughing, and eye irritation when cooking with wood but reported that symptoms were less severe when cooking with charcoal. In conclusion, we observed that exposure to PM_2.5_ was substantially reduced by cooking outdoor with charcoal. As both wood and charcoal fuels are associated with negative environmental and health impacts, the adoption of high-efficiency cookstoves and less polluting sources of energy will be highly beneficial. Cooking outside whenever possible, and minimising the time spent in close proximity to stoves, may be simple interventions that could reduce the risks of exacerbation and progression of respiratory and cardiovascular diseases in Malawi.

## 1. Introduction

The widespread combustion of biomass fuels (e.g., wood, charcoal, and crop residues) in low- and middle-income countries (LMIC) for cooking, heating, and lighting generates household air pollution (HAP), including particulate matter (PM) [1,2]. Exposure to this type of air pollution is associated with respiratory and cardiovascular diseases and has been linked to between 2.9 and 4.3 million deaths globally each year [1,3,4,5,6,7]. In an attempt to lower health risks, the World Health Organisation (WHO) has established guideline exposure limits for PM_2.5_ (PM of average aerodynamic diameters of less than or equal to 2.5 μm) of 25 µg/m^3^ and 10 µg/m^3^ for 24 h and annual averaging periods, respectively [8].

Malawi is one of the poorest countries in the world, and many of its inhabitants use biomass fuel as a supposedly cheaper way of cooking (World Bank, 2019). In common with many other developing nations, the most common cooking method in Malawi is a ‘three-stone stove’ used to burn wood [9,10,11,12,13]. The relatively few peer-reviewed studies of direct airborne particle exposure measurements in Malawi that we identified indicate high exposure concentrations [10,14,15]. Fullerton et al. [10] measured average respirable dust concentrations for 374 adults of 811 and 204 µg/m^3^ close to stoves in rural and urban areas in Malawi. Cho et al. [14] measured 366 personal 48-h PM_2.5_ exposures of children in 319 rural Malawian households ranging from 7.6 to 421.7 μg/m^3^, with an average of 49.2 μg/m^3^, with >75% participants exposed to PM_2.5_ concentrations exceeding the 25 μg/m^3^, 24-h WHO PM_2.5_ exposure guideline. Real-time personal exposures ranged from 0 to 10,850 μg/m^3^. Rylance et al. [15] measured 1768 personal exposures over monitoring periods lasting >48 h and a further 902 periods lasting between 24 and 48 h, with overall 24-h median personal PM_2.5_ exposures of 77  µg/m^3^ (interquartile range: 43–153 µg/m^3^).

Static measurements provide estimates of ambient indoor PM_2.5_ concentrations but may not represent personal exposures, as people are likely to have varying proximities to the stove area during cooking. For example, average ambient and personal PM_2.5_ exposures ranging between 33 and 940 µg/m^3^ and 34 and 522 µg/m^3^, respectively, have been measured during cookstove operations in households in Sri Lanka [16]. These observations illustrate that, even if a participant is not always at the cookstove, they can be exposed to concentrations substantially exceeding the WHO guidelines. The above studies focused mainly on indoor kitchen locations. However, from our discussions with local people in Malawi, depending on the season and climate, three-stone stoves are often used outside or in other indoor locations.

We made measurements of indoor and personal exposures to PM_2.5_ in a rural district in Malawi where both wood-fuel three-stone stoves and charcoal burners (Mbaula) are used for cooking [17]. The objectives of our research were: to use portable monitoring equipment to compare static and personal PM_2.5_ exposures associated with different cooking methods and locations; and to investigate the association between PM_2.5_ exposures and non-invasive indicators of cardiovascular and respiratory risk (blood pressure and respiratory function).

## 2. Methods

### 2.1. Sampling Location and Time

Malawi has a population of approximately 14.8 million [18], with the capital city Lilongwe located in the central region and the centre for finance and commerce Blantyre in the southern region. Our study was conducted from January to April 2017 in Kalonga village in Chikwawa District in the southwest of Malawi (Figure 1). January–April is the main rainy season in Malawi, which diminishes in April (average rainfall typically around 228 mm in January compared to around 5 mm in April). Throughout this period, the humidity generally remains high, and temperatures range from lows of around 22 °C to highs of around 32 °C.

Eight households participated in air quality monitoring, including households that cooked on either 3-stone stoves using wood or Mbaula using charcoal (Figure 2). We monitored the personal PM_2.5_ exposures of the main cooks in each household (all of whom were female; age range: 12–81). Four types of static locations were monitored: inside single-room houses, inside separate kitchen buildings, on verandas, and outside (Table 1 and Figure 3).

### 2.2. Measurement of PM_2.5_

A lightweight (<240 g) personal exposure monitor (MicroPEM (Serial no: 320763N); RTI International, Research Triangle Park, NC, USA) was used to measure PM_2.5_ concentrations throughout the deployment periods using a micro-nephelometer preceded by a size-selective impactor inlet coated with silicone oil [16]. A 1-m Tygon tube was attached to the MicroPEM inlet, and a conductive asbestos sampling inlet with a 4-inch cowl (SKC Ltd., Blandford Forum, UK) was connected to the inlet of the tubing. In previous field tests, we found that that this type of inlet configuration was effective in minimising grit and/or water ingress to the nephelometer and had a negligible effect on the measured PM_2.5_ estimates. The MicroPEM flow rate was set at 0.50 L/min. Measurements made at 10-s time intervals were averaged to 1-min intervals for the time series plots. In this study, we used the factory-calibrated MicroPEM output to estimate PM_2.5_ concentrations, which we and others have found provide reliable estimates of relative trends in elevated concentrations of PM_2.5_ [19,20,21,22,23,24]. A limitation of our study is that we were unable to correct the MicroPEM nephelometer PM_2.5_ estimates from simultaneous gravimetric measurements.

We made static measurements close to cookstoves and personal exposure measurements with the MicroPEM placed on the participant. We only had access to one MicroPEM; therefore, the static and personal exposure measurements were not made simultaneously.

For static monitoring, the MicroPEM was placed in a waterproof case (Pelicase; Peli Products UK Ltd., Glossop, UK) positioned 1 m from the cookstoves between 10 a.m. to 2 p.m. on sampling days. The 1-m Tygon tube attached to the MicroPEM inlet enabled static sampling of air outside of the protective case (Figure 4).

During cooking, it was observed that participants would often stand closer to, or bend over, the stove. With the static monitor located 1 m away from the stove, such influences on exposure would not be measured; therefore, personal exposure was monitored during separate cooking sessions in the same types of static locations. For personal monitoring, the MicroPEM was inserted into a camera bag to allow it to be carried easily and comfortably so as not to hinder the participants’ cooking activities. The 1-m Tygon inlet tube was routed from the bag and taped onto the participant’s shoulder (Figure 5). Eight participants took part in the personal monitoring experiment, one in each of the four cooking locations, using either wood or charcoal. Exposure monitoring was conducted after 10 a.m. Participants were asked not to cook earlier in the morning of the same day that personal monitoring took place to avoid exposure to combustion-related PM_2.5_ shortly before physiological measurements were made.

### 2.3. Measurement of Ambulatory Blood Pressure (BP), Heart Rate (HR) and Peak Expiratory Flow Rate (PEFR)

Four nurses from the Mfera Health Facility assisted by collecting health data throughout the cooking sessions. Participants were asked not to cook in the morning of the day personal monitoring was conducted, to minimise PM_2.5_ exposure before the experiment. We measured BP and HR with a Rossmax AW356 blood pressure monitor (Rossmax International Ltd., Heerbrugg, Switzerland) before, during, and after cooking activities. BP readings were taken in triplicate at each time point and averages calculated. Systolic BP (SBP) and diastolic BP (DBP) were compared to UK National Health Service clinical BP categories [25] (Table 2).

Similarly, PEFR was measured in triplicate five times throughout the day. The maximum PEFR from each set of triplicate readings was recorded. Normal PEFR vary by age, sex, and height; therefore, we used an online calculator to estimate the normal expected PEFR for each participant [26]. Individual PEFR were categorised using colours to represent the percentage of estimated normal value [27]. The green category represented 80–100% of the estimated normal value. The orange category represented 50–80% of the normal value, suggesting some airway narrowing. The red category represented less than 50% of the normal value, indicating severe narrowing of the airways.

### 2.4. Information on Activities and Prior Health Conditions

Each participant in the personal exposure assessment study was asked to complete a questionnaire about their activities. The questionnaire was used to gather information on the type of fuel used and breathing or eyesight difficulties while cooking. The questionnaire data complemented the field observations during each sampling period on: the type of fuel used, the number of people living in the house, and any physical symptoms related to the inhalation of smoke, e.g., breathing difficulties or eye irritation. Most participants had a health passport (official government health record containing information on the participant’s health history), which was reviewed by the nurse to inform our study of previously recorded respiratory illnesses.

## 3. Results

### 3.1. PM_2.5_ Concentrations Observed during Static Sampling

Sampling sessions ranged from 77- to 262-min durations between 11 January and 20 April 2017 (Table 1). The average PM_2.5_ concentrations from the sampling periods for cooking with wood ranged from 638 to 2184 µg/m^3^ (Table 3). In contrast, the average PM_2.5_ concentrations for cooking with charcoal ranged from 17 to 46 µg/m^3^. Ratios of the mean PM_2.5_ for wood:charcoal in different static locations ranged from 24 to 55. The maximum PM_2.5_ concentration recorded (11,733 µg/m^3^) occurred during cooking with wood on the veranda (in contrast, the highest concentration recorded with charcoal cooking on the veranda was 1378 µg/m^3^) (Table 3). Similar contrasts in the transient peak concentrations were obtained in the three other cooking locations. During cooking with wood, the maximum concentrations observed in the kitchen, house, and outside were 11,032, 11,268, and 11,242 µg/m^3^, respectively. During cooking with charcoal, the maximum concentrations observed in the kitchen, house, and outside were 251, 245, and 1707 µg/m^3^, respectively. The two lowest average concentrations (27 and 17 µg/m^3^) were recorded when cooking with charcoal on the veranda and outside, respectively. Time series plots of 1-min average PM_2.5_ concentrations indicated highly fluctuating concentrations (presumably associated with short-term cooking activities and/or air movements) in indoor locations (Figure 6). The different *y*-axis (concentration) scales on the left- and right-hand sides of Figure 6 emphasise the marked contrast in the magnitude of exposures observed between cooking with wood and charcoal.

### 3.2. PM_2.5_ Exposures Observed during Personal Sampling

The average personal PM_2.5_ exposures over individual sampling periods for cooking with wood ranged from 97 to 1163 µg/m^3^ (Table 3). In contrast, the average personal PM_2.5_ exposures for cooking with charcoal ranged from 39 to 193 µg/m^3^. Ratios of the average personal PM_2.5_ exposures for wood:charcoal in different locations ranged from 2 to 23. The lowest personal exposures were observed for outdoor cooking with charcoal. Personal exposures to PM_2.5_ in the indoor locations were 3.3 and 1.7 times greater than exposures observed in equivalent outdoor locations for wood and charcoal, respectively. For cooking with wood, the average PM_2.5_ personal exposures were lower than the average PM_2.5_ concentrations measured during static sampling, perhaps resulting from the participant not always being present in the very high-concentration microenvironment close to the wood fire throughout the cooking session. For cooking with charcoal, the average PM_2.5_ personal exposure was higher than average PM_2.5_ concentration measured during static sampling.

Very high peak exposures were associated with both fuels. The maximum personal exposure PM_2.5_ concentrations recorded when cooking with wood and charcoal were 11,370 µg/m^3^ and 10,660 µg/m^3^, respectively, with both of these peak exposures observed on the veranda (Table 3). When wood was the fuel source, the maximum personal exposure concentrations in the kitchen, in the house, and outside were 10,476; 11,296, and 9666 µg/m^3^, respectively. When charcoal was used as a fuel, the maximum personal exposure PM_2.5_ concentrations were 9164, 777, and 2328 µg/m^3^ in the kitchen, in the house, and outside, respectively. The 1-min-averaged time series showed the marked contrast between personal exposures arising from cooking with wood and charcoal (Figure 7).

### 3.3. Health Data

#### 3.3.1. Blood Pressure and Heart Rate

Many of the participants had normal BP throughout the cooking period (Table 4). However, two participants, aged 48 and 81, had very high readings, indicating hypertension. The health passport of these two participants did not provide any information concerning a history of high blood pressure, so it would be hard to determine whether this was a one-off result or not. The district nurses advised these participants to report to their nearest health facilities. There was no clear association found between short-term PM_2.5_ exposure and changes in BP or heart rate.

#### 3.3.2. Peak Expiratory Flow Rate (PEFR) Tests

Only two participants had their PEFR over 80% of the normal value (Table 5). The oldest participant had an observed PEFR below 50% of the normal age and height standardised PEFR. The remaining participants had observed PEFR between 50% and 80% of the normal PEFR. We observed reductions in the PEFR during cooking with wood in indoor locations (e.g., changes of −60 and −40 l/min in the kitchen and in the house, respectively, compared to 0 and +7 l/min in the same locations cooking with charcoal). When cooking with wood, increased PM_2.5_ exposure in this small sample of participants appeared to be associated with a greater change in the PEFR (Table 5).

#### 3.3.3. Information from Questionnaires and Health Passports

Most households indicated that they had both wood and charcoal available, depending on the time of the year and the family’s financial circumstances. Five out of eight participants (participants: 1, 3, 4, 7, and 8—Table 4) indicated breathing difficulties, coughing, and eye irritation experienced during cooking with wood and less severe symptoms during cooking with charcoal. Five out of eight participants (participants: 4, 5, 6, 7, and 8) had records in their health passports of significant Upper Respiratory Tract Infections (URTI) and respiratory discomfort; this also included younger participants. For example, the 15-year-old participant (participant 4) had had four pneumonia and four URTI episodes recorded between 2002 and 2016. The 24-year-old participant (participant 6) had recorded episodes of chest pains, headaches, and URTI between 2016 and 2017. Three participants (participants 1–3) with no illnesses recorded in their health passports had only been using their passports since 2015 or 2016.

## 4. Discussion

The limited sample size (i.e., eight microenvironments and eight participants with non-repeated measurements) is an important limitation of our study. Therefore, we emphasise that our interpretation of the data collected in this pilot study has been done in a descriptive hypothesis-generating manner rather than using inferential statistical methods to test the hypotheses. Allowing for the above limitation, our highly detailed descriptive study highlights the likelihood of very high PM_2.5_ exposures in a remote rural community that would otherwise have few, if any, air pollution exposure measurements and demonstrates the usefulness of miniature battery-operated monitoring technology that could be deployed more extensively for hypothesis testing in larger-scale studies.

We compared two main types of fuel (charcoal and wood) in four types of cooking locations (inside houses, inside kitchen buildings, on a veranda, and outside locations). We observed large differences between particulate pollution exposure associated with different fuel types, between static and personal monitoring, and between cooking locations (especially between indoor and outdoor microenvironments).

During static monitoring, wood and charcoal combustion resulted in average PM_2.5_ concentrations ranging from 638 to 2184 µg/m^3^ and 17 to 47 µg/m^3^, respectively (Table 3 and Figure 6). The large difference between PM_2.5_ concentrations associated with wood combustion compared to charcoal combustion are consistent with the relative rankings of these fuels in earlier research [2,28,29].

Our observations can also be compared to similar static PM_2.5_ measurements using MicroPEM monitors in 11 Sri Lankan households with traditional wood-fuel cookstoves without chimneys (a design similar to the three-stone stove used in Malawi) that ranged between 37 and 940 µg/m^3^ (average: 369 µg/m^3^) [16,30]. In the Sri Lankan study, static PM_2.5_ measurements were made over 48-h periods by placing MicroPEM monitors 1.5 m from the ground level and 1.5 m from the cookstove. The combination of longer measurement periods, including periods when cooking would not have been taking place, together with slightly greater vertical and horizontal distances between the monitor and cookstove may explain the lower PM_2.5_ concentrations measured in the Sri Lankan study compared to our measurements. Other cookstove studies in Africa and Asia have reported PM_2.5_ concentrations over 1000 µg/m^3^ [10,31].

Static measurements allow an initial estimate of the potential exposure to PM_2.5_ concentrations at fixed locations close to cooking activities but do not represent what the cook was actually exposed to, as they would be moving around at different distances from the stove during cooking. Personal monitoring, where the participant carries the monitoring instrument, enables a more direct estimation of personal exposure to PM_2.5_ over the sampling period. MicroPEMs are ideal for personal exposure monitoring, as they are lightweight, allowing them to be easily carried in a comfortable manner. In our study, personal monitoring confirmed the observations during static sampling that wood produced higher PM_2.5_ concentrations than charcoal. When cooking with wood, the participant was exposed to 1-min average PM_2.5_ concentrations as high as 7000 µg/m^3^ with wood compared to 3600 µg/m^3^ using charcoal (Figure 7), equating to 280 and 144 times greater than the WHO 24-h guideline value of 25 µg/m^3^. Our observations (e.g., average personal exposure to wood smoke of 706 µg/m^3^) can be compared to the average personal PM_2.5_ wood smoke exposures in the comparable Sri Lankan MicroPEM monitoring study of 34–522 µg/m^3^ [16,30]. Analogous to the comparisons made for the static exposures above, our personal exposure measurements were higher than the equivalent measurements in Sri Lanka, which may be the result of the latter being made over 48-h periods, including extended periods without cooking activities.

We observed that the concentrations measured during personal exposure measurements for cooking with charcoal were consistently higher than the static measurements for cooking with charcoal in the same types of locations (Table 3). This might have been the result of the participants bending over the charcoal cookstoves, and therefore, the MicroPEM recorded transient concentrations that were far higher than when the instrument was placed 1 m away from the cookstove in static measurements (Table 3). However, when wood was used for cooking, both the average and peak personal exposures were generally lower than the concentrations observed by static measurements. Perhaps this was the result of greater amounts of smoke deterring the participants from bending closer to the wood-fuel cookstoves, coupled with time periods at greater distances from the cookstove. Lower personal vs. static exposures have been observed in other research studies [16,29].

In addition to the measurement of PM_2.5_ associated with different cooking fuels, the effect of the cooking location was also assessed. Both static and personal sampling consistently resulted in higher PM_2.5_ concentrations in indoor locations (i.e., inside the house and kitchen buildings), compared to outdoor locations (i.e., on the veranda and outside). For example, for cooking with wood outside, the average PM_2.5_ was 929 µg/m^3^, compared to cooking with wood inside the house when the average PM_2.5_ was 2184 µg/m^3^. In outdoor microenvironments, airborne particles released by cooking disperse more readily as a result of increased air movement and fewer physical barriers to dispersion. Our observations are consistent with other research comparing indoor and outdoor pollution exposure in rural locations in Africa [29].

Even though the average static PM_2.5_ concentrations measured for cooking with wood in indoor environments (i.e., in the single-room house or in separate kitchen building) were higher than for cooking on the veranda or outside, transient peak static PM_2.5_ concentrations exceeding 11,000 µg/m^3^ were observed in all locations. These very high transient concentrations may have resulted from short-term air movements in close proximity to stoves advecting relatively undiluted combustion plumes directly towards and around the PM_2.5_ monitor. Similarly, very high peak concentrations for personal exposure measurements were observed in all locations, especially for cooking with charcoal, perhaps (as we discuss above) as a result of participants bending over the stove for short time intervals. Further research could examine these effects in more detail, including analyses of ventilation characteristics of each type of location. Fullerton et al. 2009 described the use of similar cooking locations in earlier research in Malawi and, similar to our observations, noted that most participants cooked indoors, especially during the wet season.

Our study emphasised the extent of the very high PM_2.5_ concentrations measured in both static and personal monitoring. Although we showed that cooking with charcoal produced less PM_2.5_ than wood, the average personal exposure was substantial no matter which fuel or cooking location was used. All of our average PM_2.5_ measurements were markedly higher than the guidelines set by the WHO of 25 µg/m^3^ and 10 µg/m^3^ for 24-h and annual averaging periods, respectively [8]. In other words, all of our observations indicated PM_2.5_ concentrations at which detrimental health effects can occur through prolonged exposure.

Emissions from biomass fuel are widely recognised as a major health concern and have been associated extensively with pulmonary and cardiovascular diseases [7,32]. Our study observed breathing difficulties amongst the participants, with some participants indicating more difficulties when cooking with wood, which is consistent with research on a larger population sample in Malawi [33]. Based on the health passports, many breathing illnesses were recorded for the participants, some repeatedly. A detailed systematic review on COPD associated with biomass fuel use in women confirmed that exposure to biomass smoke is associated with COPD [6]. Our research aligned with the findings of this review, as five out of eight participants experienced repeated pulmonary illnesses. In addition to the participant cooks, a number of people, including infants and very young children, also gathered near the cookstove for different amounts of time during the sampling session. These individuals would have been exposed to similar pollution concentrations as the cook and, therefore, may have been similarly at risk of developing lung diseases. Infants may be at increased risk due to their developing respiratory systems and small stature and, hence, a closer proximity to the pollution source [34].

Although numerous studies have investigated the associations between exposure to biomass pollutants and coronary heart disease (e.g., [35]), a relatively small number of studies have examined exposure to HAP and changes in blood pressure [5,36,37]. We monitored SBP and DBP before, during, and after cooking sessions. We observed small (not always with consistent direction) variations in SBP and DBP throughout the day. Changes in SBP were mostly negative, ranging between −27 and +3 mmHg during the cooking sessions (Table 4). Norris et al. 2016 observed SBP decreases by −0.4 to −0.2 mmHg in a study of women with exposure to cookstove emissions in rural communities in India. Our relatively small sample size and short period of exposure measurements may have obscured associations between air pollution exposure and BP over longer timescales [38,39].

## 5. Conclusions

We measured static and personal exposure to PM_2.5_ in four types of cooking locations and compared wood and charcoal as the cooking fuels. The static air monitoring showed that the charcoal stoves produced substantially less PM_2.5_ than three-stones stoves with wood fuel, although it is appreciated that there are substantial environmental and human costs associated with the use of charcoal fuel [40]. Correspondingly, the adoption of high-efficiency cookstoves and less polluting sources of energy will be highly beneficial [41]. Cooking outside reduced the PM_2.5_ concentrations through the dispersion of airborne pollutants. In contrast, indoor cooking generated very high PM_2.5_ concentrations and correspondingly substantial risks to human health. It would appear highly beneficial to examine ways of encouraging outdoor cooking, including steps to remove potential barriers to behavioural change. For example, it may be possible to use simple engineering interventions and/or education to provide people with the capability to construct safe, reliable, and low-cost structures to allow cooking under a ventilated canopy to avoid problems with rainfall interfering with cooking. For cooking with wood, the personal exposures were lower than the static measurements of PM_2.5_, suggesting that minimising the distance spent in close proximity to stoves may be another simple and effective intervention to reduce exposure. We noted tentative evidence of an exposure–response relationship for the association between PM_2.5_ and short-term reductions in PEFR, albeit in a very small sample, with possible confounding by the age of participants. Our analyses of the associations between PM_2.5_ and BP were inconclusive and possibly obscured by the small sample size and short exposure periods. Individual health records and reported symptoms suggested that participants were afflicted by substantial health burdens that may plausibly be associated with the very high PM_2.5_ exposures that we observed during field measurements, emphasising the potential benefits of simple low-cost interventions to strategically improve living conditions.

## Figures and Tables

**Figure 1 ijerph-18-07680-f001:**
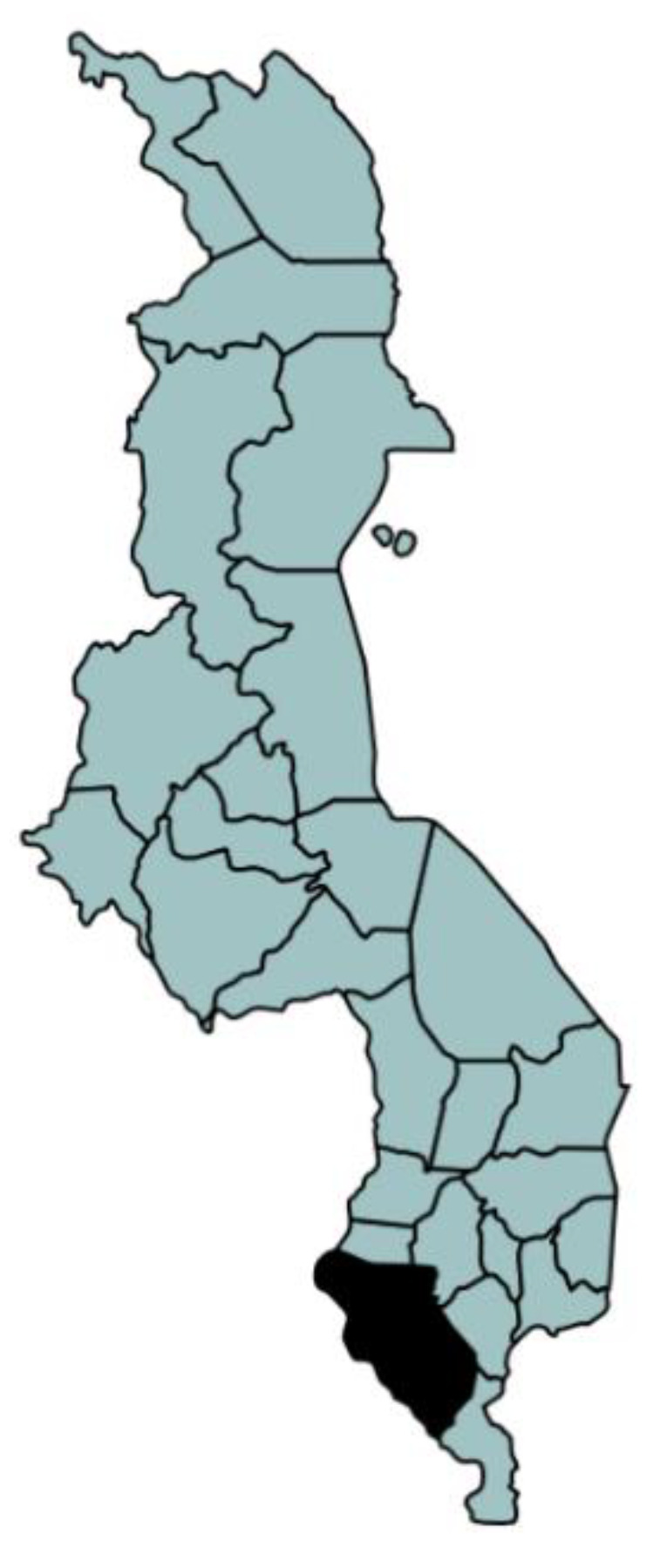
Geographical location of the Chikwawa District study area in the southwest of Malawi (darkened area on the map).

**Figure 2 ijerph-18-07680-f002:**
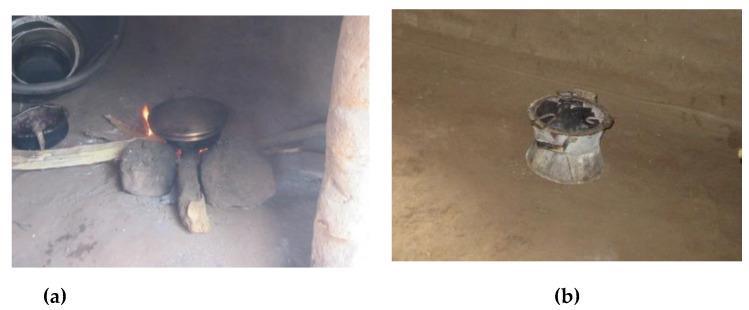
Two types of cookstoves used in Kalonga Village, Chikwawa: a three-stone stove with wood (**a**) and Mbaula charcoal burner (**b**).

**Figure 3 ijerph-18-07680-f003:**
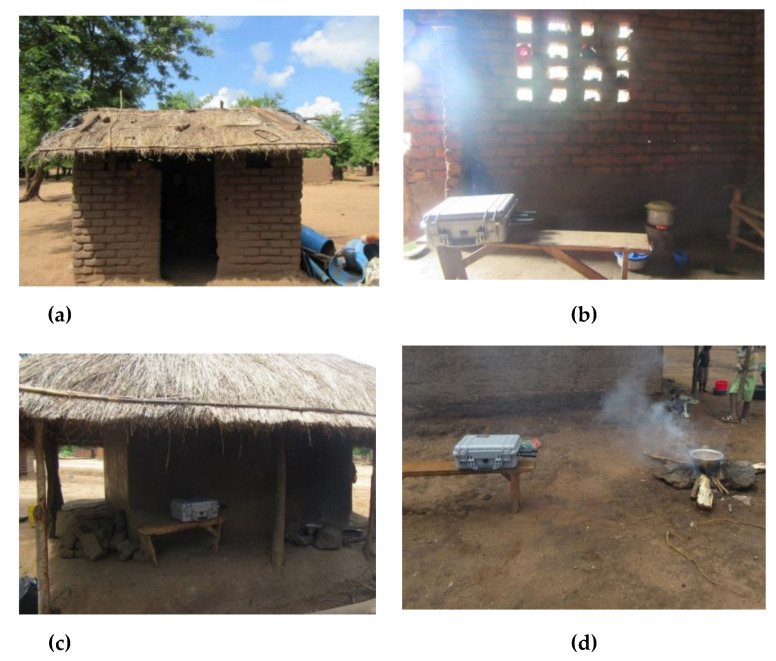
Examples of the four types of cooking locations in Kalonga Village. Separate kitchen (**a**), inside single-room house (**b**), on veranda of the house (**c**), and outside (**d**).

**Figure 4 ijerph-18-07680-f004:**
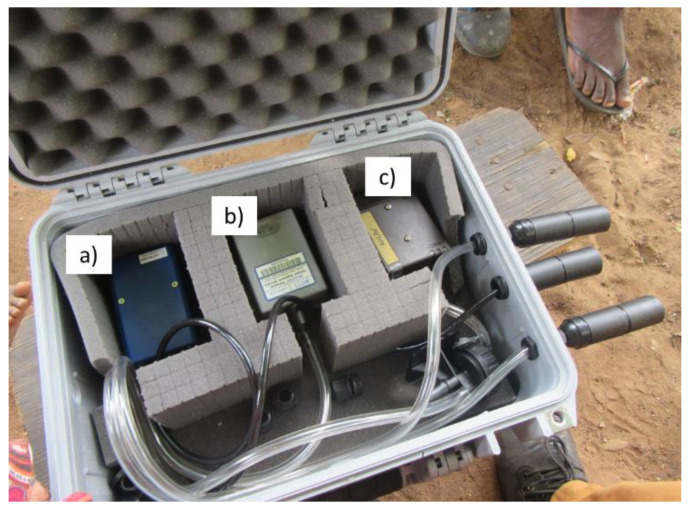
Equipment deployed in Pelicase: (**a**) MicroAeth for Black Carbon measurement, (**b**) MicroPEM for PM_2.5_ measurement, and (**c**) Airlite sampling pump for respirable dust measurement. This paper focuses on MicroPEM PM_2.5_ measurements.

**Figure 5 ijerph-18-07680-f005:**
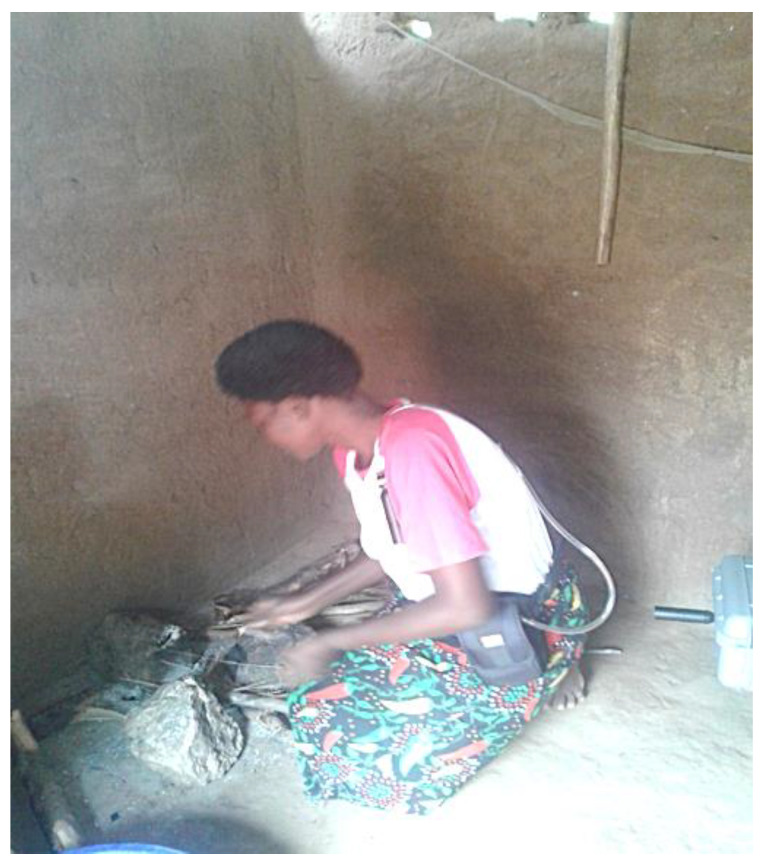
Participant carrying a MicroPEM instrument for PM_2.5_ personal exposure monitoring.

**Figure 6 ijerph-18-07680-f006:**
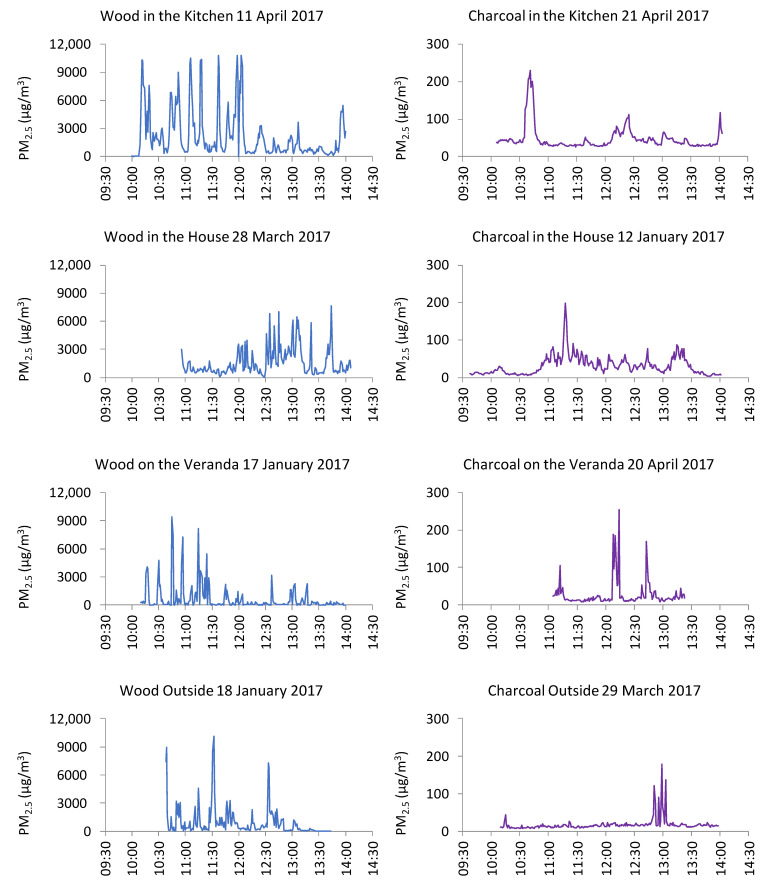
One-minute-average PM_2.5_ concentrations from static sampling in the four location types using both wood cookstoves (**left hand side of the figure**) and charcoal cookstoves (**right hand side of the figure**). N.B. The scale on the *y*-axis for cooking with charcoal cookstove graphs (**right-hand side**) is 40 times smaller than the *y*-axis for wood cookstove graphs (**left-hand side**).

**Figure 7 ijerph-18-07680-f007:**
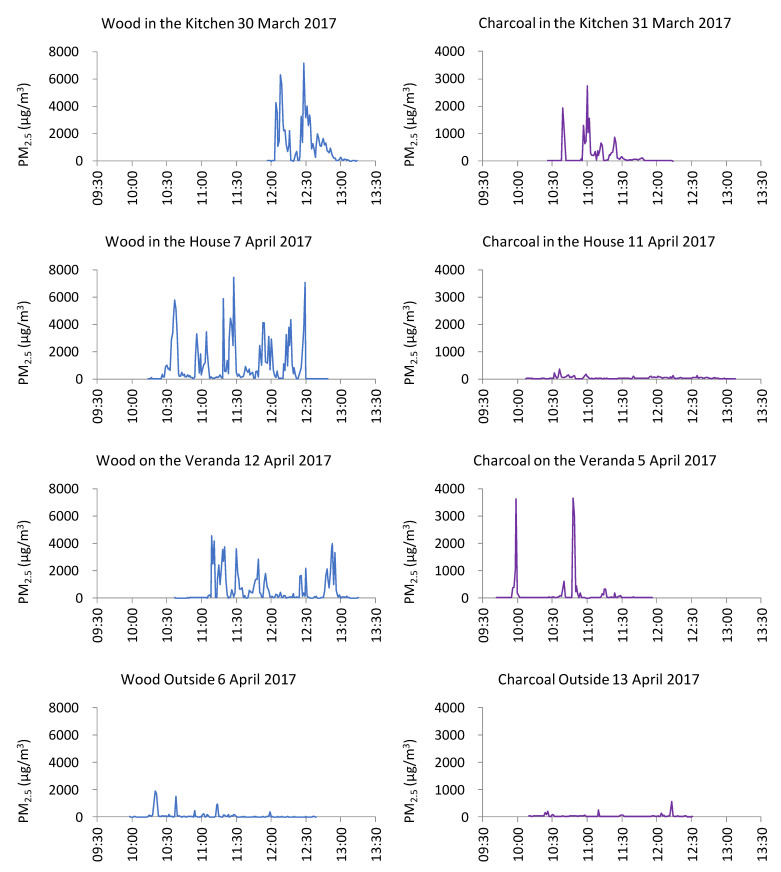
One-minute-average PM_2.5_ concentrations from personal exposure sampling during cooking in different locations: wood cookstoves (**left hand side of the figure**) and charcoal cookstoves (**right hand side of the figure**). N.B. The scale on the *y*-axis for charcoal cookstoves graphs (**right-hand side**) is half the scale of the *y*-axis for wood cookstoves graphs (**left-hand side**).

**Table 1 ijerph-18-07680-t001:** Summary of the sampling dates [dd/mm/yy] (and durations) for each sampling location and fuel type.

Location	Static		Personal	
	Wood	Charcoal	Wood	Charcoal
Kitchen	11/01/17	21/04/17	30/03/17	31/03/17
	(210 min)	(231 min)	(77 min)	(108 min)
House	28/03/17	12/01/17	07/04/17	11/04/17
	(190 min)	(262 min)	(155 min)	(181 min)
Veranda	17/01/17	20/04/17	12/04/17	05/04/17
	(228 min)	(138 min)	(158 min)	(134 min)
Outside	18/01/17	29/03/17	06/04/17	13/04/17
	(185 min)	(227 min)	(161 min)	(141 min)

**Table 2 ijerph-18-07680-t002:** Blood pressure categories (NHS 2018). Colour coding represents normal (green), elevated (orange), and high (red) blood pressure categories.

Systolic Blood Pressure (mmHg)	Diastolic Blood Pressure (mmHg)	Category
90–120	60–80	Normal
120–139	80–89	Elevated
>140	>90	High

**Table 3 ijerph-18-07680-t003:** Summary of the observed mean and maximum PM_2.5_ concentrations for each sampling period. The table also gives wood:charcoal (W:C) PM_2.5_ ratios for each location and sampling type and indoor:outdoor (In:Out) ratios for the grouped indoor and outdoor measurements.

Statistic:	Location:	Static	Static	Personal	Personal	Static	Personal
		Wood	Charcoal	Wood	Charcoal	W:C Ratio	W:C Ratio
Mean:	Kitchen	2184	46	1163	193	47	6
	House	1602	33	1008	44	49	23
	Veranda	638	27	554	100	24	6
	Outside	929	17	97	39	55	2
Mean:	Indoor	1893	40	1086	119	48	9
	Outdoor	784	22	326	70	36	4
	All	1338	31	706	94	44	7
	In:Out ratio	2.4	1.8	3.3	1.7		
Max:	Kitchen	11,032	251	10,476	9164	44	1
	House	11,268	245	11,296	777	46	15
	Veranda	11,733	1378	11,370	10,660	9	1
	Outside	11,242	1707	9666	2328	7	4
Max:	Indoor	11,150	248	10,886	4971	45	2
	Outdoor	11,488	1543	10,518	6494	7	2
	All	11,319	895	10,702	5732	13	2
	In:Out ratio	1.0	0.2	1.0	0.8		

**Table 4 ijerph-18-07680-t004:** Blood pressure (BP) and heart rate measurements before, during, and after personal PM_2.5_ exposure monitoring. Colour coding represents normal (green), elevated (orange), and high (red) blood pressure categories consistent with the colour coding in Table 2.

					Systolic BP (mm Hg)				Diastolic BP (mm Hg)				Heart Rate (bpm)			
Fuel	Location	Participant	Age	Height (cm)	Before	During	After	Difference	Before	During	After	Difference	Before	During	After	Difference
Wood	Kitchen	1	30	163	116	118	109	−7	73	75	71	−2	106	108	101	−5
Wood	House	2	81	154	150	123	123	−27	92	75	75	−17	91	81	81	−10
Wood	Veranda	3	48	162	158	159	152	−6	82	86	76	−6	81	75	79	−2
Wood	Outside	4	15	148	120	111	114	−6	73	69	68	−5	137	123	113	−24
Charcoal	Kitchen	5	12	160	104	103	95	−9	72	72	71	−1	84	82	87	3
Charcoal	House	6	24	166	113	97	108	−5	76	65	72	−4	64	79	77	13
Charcoal	Veranda	7	33	155	103	106	106	3	73	71	75	2	78	80	77	−1
Charcoal	Outside	8	21	160	110	105	104	−6	79	75	74	−5	83	84	86	3

**Table 5 ijerph-18-07680-t005:** Peak Expiratory Flow Rate (PEFR) measurements during personal sampling cooking periods compared (as %) to normal PEFR calculated with a Clement Clarke International PEFR calculator [26]. PEFR was measured from the start (Obs 1) to the end (Obs 5) of the cooking period, with Obs 2–4 as evenly spaced as possible within the cooking period. The change in PEFR is the difference in the PEFR between the first and last measurements. Green text represents participant PEFRs between 80% and 100% of the normal PEFR. Orange text represents participant PEFRs between 50% and 80% of the normal PEFR. Red text represents participant PEFRs below 50% of the normal PEFR adjusted for the age, sex, and height of the individual participants [27].

Fuel/	Age	Height	Normal PEFR	Obs 1 PEFR	Obs 2 PEFR	Obs 3 PEFR	Obs 4 PEFR	Obs 5 PEFR	Change in PEFR
Location	(years)	(cm)	(L/min)	(L/min)	(Lmin)	(L/min)	(L/min)	(L/min)	(L/min)
Wood:									
Kitchen	30	163	441	450 (102%)	500 (113%)	450 (102%)	450 (102%)	390 (88%)	−60 (−14%)
House	81	154	319	180 (56%)	160 (50%)	200 (63%)	190 (60%)	140 (44%)	−40 (−13%)
Veranda	48	162	418	350 (84%)	340 (81%)	330 (79%)	280 (67%)	300 (72%)	−50 (−12%)
Outside	15	148	380	220 (58%)	240 (63%)	250 (66%)	240 (63%)	230 (61%)	10 (3%)
Charcoal:									
Kitchen	12	160	394	240 (61%)	240 (61%)	290 (74%)	260 (66%)	240 (61%)	0 (0%)
House	24	166	438	248 (57%)	270 (62%)	269 (61%)	225 (51%)	255 (58%)	7 (2%)
Veranda	33	155	431	310 (72%)	300 (70%)	230 (53%)	270 (63%)	240 (56%)	−70 (−16%)
Outside	21	160	423	250 (59%)	290 (69%)	240 (57%)	250 (59%)	260 (61%)	10 (2%)

## Data Availability

The research data associated with this paper are available at: https://doi.org/10.15129/0330e8f8-343a-47e0-a6ad-7d0abf837720 (Accessed on 12 July 2021).

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
