# Peer review of "Exposure to Air Pollution in Rural Malawi: Impact of Cooking Methods on Blood Pressure and Peak Expiratory Flow"

_ijerph, 2021, doi:10.3390/ijerph18147680_

Round 1
Reviewer 1 Report
While I appreciate this study, conducted with limited resources where sampling equipment may not be as available due to various reasons, I have some concerns and the following major comments.
- My main concern is the limited number of samples taken. With only 8 individuals measured, and not repeated measures, the conclusions that can be arrived at with confidence are narrow. It is not possible to do any statistics to show associations or differences. As a pilot study showing PM concentrations in various situations it is ok, however, with such a variable age group and limited samples, I don't think the health data can be associated with any PM exposures with confidence.
- The methods section needs to improve. While the sampling methods and locations are clearly explained, I think there should be a paragraph giving information on the number of participants, age etc (basically study participant characteristics).
- In results - how were the associations between BP and PEF and PM analyzed. Was it just the percentage change vs. PM concentration, or were any statistical analyses done?
- The limitations of the study are not discussed adequately in the paper. I think given the small number of individuals sampled and only one measurement taken at a time, these limitations need to be discussed in detail.
Reviewer 2 Report
The manuscript titled "Exposure to air pollution in rural Malawi: impact of cooking methods on blood pressure and peak expiratory flow" presents an interesting analysis of PM2.5 exposure with both environmental and personal exposure in relation to cooking habits. The topic is important from a publication health perspective, despite the limited number of samples.
The manuscript is generally well written. Some revisions are suggested to improve the good quality of the report.
Tables must be numbered according the order they are cited in the text. Please check and revise.
The circles indicated in Figure 1 are not clear which they are. It seems all the cities in the map. Is that correct? Also please clarify the meaning of the star
I would use the term sex in lieu of gender since biological and not behavioural differences are assessed.
The association with BP is very hard to identify considering the limited number of subjects and plausible confounding of age, thus I would tone down the relation with PM2.5. Similar confounding factors is present for PEFR.
Strengths and limitations are missing. Please add them.
Reviewer 3 Report
In manuscript ID: ijerph-1267529, entitled "Exposure to air pollution in rural Malawi: impact of cooking methods on blood pressure and peak expiratory flow", the authors aim to use portable monitoring instrumentation to compare fixed and personal PM2.5 exposures associated with different cooking methods and locations.
The work may be of interest, but surely the discussions should be implemented. My major concern regards the low number of subjects investigated (only 8), which makes the results obtained not solid from a statistical point of view, especially if the results relating to PM2.5 exposure are associated in some way with health outcomes (blood pressure and peak expiratory flow). Detailed comments are reported below:
Abstract:
- I believe that some more information should be reported here, regarding the methods used in the study.
Introduction:
- Lines 38-41: I suggest adding some references.
- Lines 54-56: I believe that this sentence is not well contextualized in the text. In addition, are there other similar (and more recent) studies?
- Lines 65-67: I think this passage is very interesting and perhaps it should be explored better.
Methods:
- Line 82: Can the authors add some more information regarding the weather conditions present during the monitoring campaign?
- Line 82: Eight households only? I believe that the number is really limited: I suggest discussing this issue, perhaps in the conclusions of the study.
- Line 86: I believe the authors got confused with the numbering of the tables in the text: I suggest checking and placing the tables/figures next to the reference text (in this case table 4 should be numbered as 1, correct?).
- Figure 1: I'm sorry but I can't see the circle referenced in the caption: if there is, I suggest making it more evident ?.
- Figure 2: Right/left become top/bottom? To avoid confusion, I suggest indicating the two images as (a) and (b).
- Line 96 - Table 4: I believe that before reporting this table it is necessary to indicate which methods have been used for the measurement of blood pressure and heart rate. In addition, I think it is appropriate to report the meaning of the colors in this caption as well.
- Line 102-Table 1: I suggest also reporting the monitoring time for the different cases in the table.
- Figure 3: Also in this case I suggest an identification of the figures as (a), (b), (c), (d).
- Line 110: Was the PM2.5 data corrected in any way after analysis?
- Line 146: Even if BC will not be considered in this paper, I still suggest reporting the BC definition.
- Line 155: Perhaps it is not well specified in the text in which period the monitoring was performed: I suggest specifying it better in the text.
Results:
- Line 210-Table 2: If I am not mistaken, the text does not mention the different ratios shown in this table. If it's not even mentioned, I suggest deleting this information from the table.
- Line 217-figure 6: In line 187 the authors report that “Sampling sessions ranged from 22 to 72 minutes […]” but from these graphs it seems that the monitoring lasted much longer. Can the authors be more specific about it?
- Line 248 - Figure 7: So there is only one sample per type of location/type of cooking? This aspect should really be discussed better.
- Line 261: How was this association evaluated? With statistical tests??
- Line 266: Why was an exclusion criterion not used, for example, based on age? If these were chosen for a reason (e.g., increased variability), this should be better defined.
- Line 282- Table 5: With reference to Obs1-5: How often were these evaluations carried out? If I'm not mistaken, it is not well specified in the text.
- Paragraph 3.3.3: Can the authors be more specific when they talk, for example, about: “five out of eight participant indicates […]”: who are these subjects? The older ones? I suggest referring to a table with this information.
Discussions:
- I believe that this section must necessarily be implemented as there are no real discussions regarding the results obtained.
Round 2
Reviewer 1 Report
The authors have improved the manuscript greatly by addressing the main issues. I have the following comments with regards to improving the manuscript.
- Abstract: Given the small sample size, I don't believe conclusions can be drawn on statistical associations between PM exposure and BP and PEFR. Therefore, I suggest changing lines 15-16, 22-23 and 25-26 to not draw definite conclusions. For example "There was no clear association found between short term PM2.5 exposure and BP" cannot be concluded because other factors may have influenced this association. May be say "..although out sample size was small, we observed...." and then explain.
Author Response
Thank you for this advice. We have amended the Abstract accordingly.
Reviewer 3 Report
The authors provided complete answers and implemented all suggested changes. The revised manuscript appears to be significantly improved over the original version. I have no other comments to submit
Author Response
Thank you for this feedback.